# BioAssay templates for the semantic web

Alex M. Clark, Nadia K. Litterman, Janice E. Kranz, Peter Gund, Kellan Gregory and Barry A. Bunin

Collaborative Drug Discovery, Inc., Burlingame, CA, United States of America



## ABSTRACT

Annotation of bioassay protocols using semantic web vocabulary is a way to make experiment descriptions machine-readable. Protocols are communicated using concise scientific English, which precludes most kinds of analysis by software algorithms. Given the availability of a sufficiently expressive ontology, some or all of the pertinent information can be captured by asserting a series of facts, expressed as semantic web triples (subject, predicate, object). With appropriate annotation, assays can be searched, clustered, tagged and evaluated in a multitude of ways, analogous to other segments of drug discovery informatics. The BioAssay Ontology (BAO) has been previously designed for this express purpose, and provides a layered hierarchy of meaningful terms which can be linked to. Currently the biggest challenge is the issue of content creation: scientists cannot be expected to use the BAO effectively without having access to software tools that make it straightforward to use the vocabulary in a canonical way. We have sought to remove this barrier by: (1) defining a BioAssay Template (BAT) data model; (2) creating a software tool for experts to create or modify templates to suit their needs; and (3) designing a common assay template (CAT) to leverage the most value from the BAO terms. The CAT was carefully assembled by biologists in order to find a balance between the maximum amount of information captured vs. low degrees of freedom in order to keep the user experience as simple as possible. The data format that we use for describing templates and corresponding annotations is the native format of the semantic web (RDF triples), and we demonstrate some of the ways that generated content can be meaningfully queried using the SPARQL language. We have made all of these materials available as open source (http://github.com/cdd/bioassay-template), in order to encourage community input and use within diverse projects, including but not limited to our own commercial electronic lab notebook products.

Corresponding author
Alex M. Clark,
aclark.xyz@gmail.com

## INTRODUCTION

One of the major problems currently being faced by biologists charged with the task of performing experimental assays on pharmaceutically interesting molecules is the information burden involved with handling collections of assay descriptions. Individual laboratories may carry out hundreds or even thousands of screening experiments each year. Each of these experiments involves a protocol, and any two experiments may be identical, similar, or completely different. The typical practice for describing bioassay protocols, for both external communication and internal record keeping,

is to use concise scientific English, which is the most universally human readable method of communication, assuming the recipient is familiar with the relevant jargon.

Unfortunately this method is not scalable. Even given the availability of an expert, it is often quite difficult and time-consuming to read two assay description paragraphs and provide a metric for the degree to which two protocols differ. There are many workflow scenarios where comparison of protocols is necessary, e.g. searching through a collection of previous experiments, or making a judgment call as to whether two batches of small molecule measurements are comparable. Attempting to use software to assist with such tasks, when the substrate is unconstrained text, results in solutions that are crude at best.

While these issues with scalability could be described as a relatively minor nuisance in a small laboratory, the field of drug discovery has lately been undergoing a renaissance of open data (*Clark, Williams & Ekins, 2015*; *Hersey, Senger & Overington, 2012*; *Ecker & Williams-Jones, 2012*; *Williams, Wilbanks & Ekins, 2012*). Services such as PubChem provide a truly massive resource (*Helal et al., 2016*); PubChem alone provides more than a million unique bioassay descriptions, and is growing rapidly (*Bolton, 2015*).[1] Such data are supplemented by carefully curated resources like ChEMBL (*Gaulton et al., 2012*), which are much smaller but have strict quality control mechanisms in place. What these services have in common is that their bioassay protocols have very little machine-readable content. In many cases, information about the target, and the kind and units of the measurements, have been abstracted out and represented in a marked up format, but all of the remaining particulars of the protocol are ensconced within English grammar, if at all.

In order to address this problem, the BioAssay Ontology (BAO) was devised (*Abeyruwan et al., 2014*; *Vempati et al., 2012*).[2] The BAO, which includes relevant components from other ontologies, is a semantic web vocabulary that contains thousands of terms for biological assay screening concepts, arranged in a series of layered class hierarchies. The BAO is extensive and detailed, and easily extensible. The vocabulary is sufficiently expressive to be used for describing biological assays in a systematic way, yet it has seen limited use. Influential projects such as PubChem (*Kim et al., 2016*), ChEMBL (*Willighagen et al., 2013*), BARD (*de Souza et al., 2014*) and OpenPHACTS (*Williams et al., 2012*) make use of the ontology, but the level of description in each is shallow, using only a small fraction of the terms.

There are a number of factors holding back scientists from using the BAO and related ontologies to describe their assays in detail, with perhaps the most substantial being the lack of software that makes the annotation process fast and convenient. Because it is based on the semantic web, BAO concepts are expressed as triples, of the form [*subject*, *predicate*, *object*]. There are no hard rules about how this is applied, which is a characteristic of the semantic web, and is both an asset and a liability. The simplest way to consider annotating a particular feature of an assay, e.g. the biological process, is to compose a triple of a form such as [*assay ID*, *biological process*, *viral genome replication*]. Each of these three fields is a uniform resource indicator (URI), which points to a globally unique object with established meaning. In this case, *assay ID* would

[1] It should be noted that the majority of the first million PubChem assays do not contain detailed experimental assay descriptions. Contributors such as the Broad Institute and organizations affiliated with the Molecular Libraries Screening Center can be selected by browsing the sources: http://pubchem.ncbi.nlm.nih.gov/sources/sources.cgi.

[2] The materials for the BAO can be found at http://bioassayontology.org.

correspond to an identifier that the user has created for the assay description; *biological process* corresponds to a specific property in the BAO that is used to link assays and the biological process that is being affected; and *viral genome replication* refers to a class in the BAO, which identifies a specific instance of a biological process, which is in turn inherited from a sequence of increasingly general classes, and may also be linked to any other node within the greater semantic web, such as the extensive Gene Ontology (GO) (*The Gene Ontology Consortium, 2015*).

In principle, screening biologists can use the properties and classes from the BAO to annotate their assays intelligently in a machine readable format that is compatible with the universe of the semantic web. If large numbers of assays were sufficiently annotated, biologists and other drug discovery scientists could perform advanced searches and filtering that would enable better interpretation of results, enhanced building of machine-learning models, and uncovering of experimental artifacts. Despite the clear benefits of semantic annotation, the BAO remains largely unused, the primary reason being its lack of accessibility. The BAO and its linked dependencies are large, and can be expected to keep growing as they are extended to capture more biological concepts. For an interactive view onto these terms, the site http://bioportal. bioontology.org/ontologies/BAO should be used to peruse the hierarchy.[3] Figure 1 shows two snapshots of part of the BAO hierarchy, using the BioPortal resource. The *classes* (Fig. 1A) that make up the ontology contain the bulk of the terms and provide most of the expressive value, while the *properties* (Fig. 1B) are used to provide context. The class hierarchy is in places many levels deep, and although it is arranged in a logical pattern, it is nonetheless necessary to be familiar with the entire layout in order to meaningfully annotate an assay protocol. Even an expert biologist familiar with the entire ontology would be presented with multiple degrees of freedom for deciding how to annotate a protocol; this is a fundamental problem for machine readability, which requires uniform consistency.

In our previous work we addressed the end-user problem, and invented technology that applies to the scenario when a user is presented with plain English text, and is charged with the task of selecting the appropriate semantic annotations. Our solution involved a hybrid approach that combined natural language processing with machine learning based on training data, with an intuitive interface that helps the user select the correct annotations, leaving the final choice in the hands of the scientist (*Clark et al., 2014*). During this process we found that the challenge that we were unable to fully overcome was the burden of creating new training data. The BAO vocabulary defines more than 2,500 classes, in addition to properties and terms from other ontologies, all of which can be expected to grow as the BAO is increasingly used for more biological content.

Considering each term as it applies to a given assay requires a high level of expertise of the BAO itself. For example, the NIH's Molecular Libraries Program's bioassay database, known as the BARD, employed dedicated research staff to annotate more than two thousand assays (*de Souza et al., 2014*). The absence of clear and straightforward guidance as to which terms to use under what circumstances is preventing adoption of the BAO

[3] It can also be browsed and edited using software such as Protégé, which can be found at http://protege.stanford.edu.

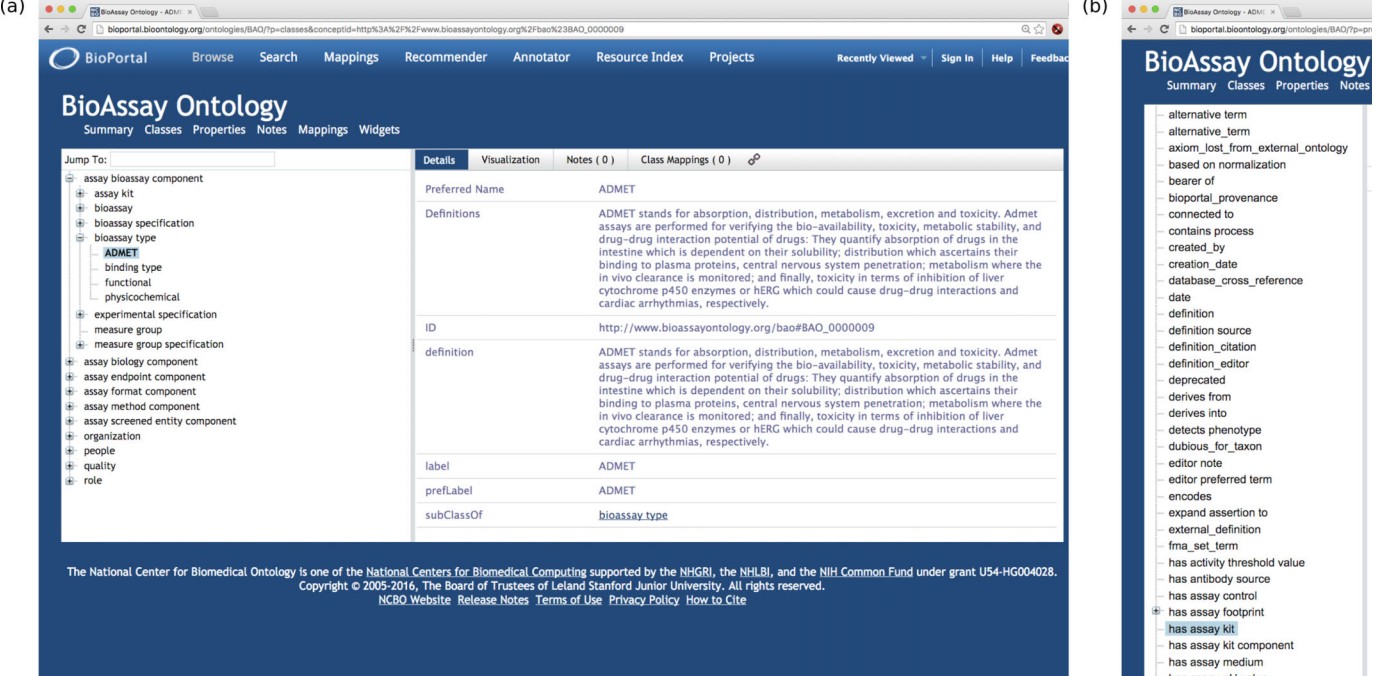

**Figure 1** A selection of the BAO hierarchy, visualized using BioPortal (http://bioportal.bioontology.org). (A) classes and (B) properties.

by drug discovery scientists. For our model building efforts, we made use of a training data set made up of 1066 PubChem bioassays that each had more than a hundred terms associated with them (*Wang et al., 2014*; *Schürer et al., 2011*), although not all of the annotations were able to be matched to ontology terms. For purposes of creating additional training data, we experienced considerable difficulty finding what we considered to be canonical annotations for any given assay.

The BAO is essentially a vocabulary that is capable of describing many assay properties, but it lacks instructions on its use. This is an issue that we have undertaken to solve, and in this article we describe our approach to providing this critical missing component.

We describe a data model called the BioAssay Template (BAT), which consists of a small number of terms which are organized to describe *how* the BAO and linked ontologies should be used to describe a particular kind of bioassay. A template is essentially a gateway to the overall ontology, which divides the assay annotation process into a fixed hierarchy of *assignments*, each of which has a prescribed list of *values*, which are cherry-picked from the overall ontology.

The BAT vocabulary can be used to create any number of templates, which can be customized to suit the task at hand. As a starting point, we have created what we refer to as the *common assay template* (CAT). CAT is an annotation recipe that is intended to capture the major properties that most biologists need to describe their assays and that enables most drug discovery scientists to have a basic understanding of an assay and its results.

A condensed summary of this template is shown in Fig. 2. Unlike the class hierarchy of the BAO, the tree structure of the CAT is flat. While the data model allows groups

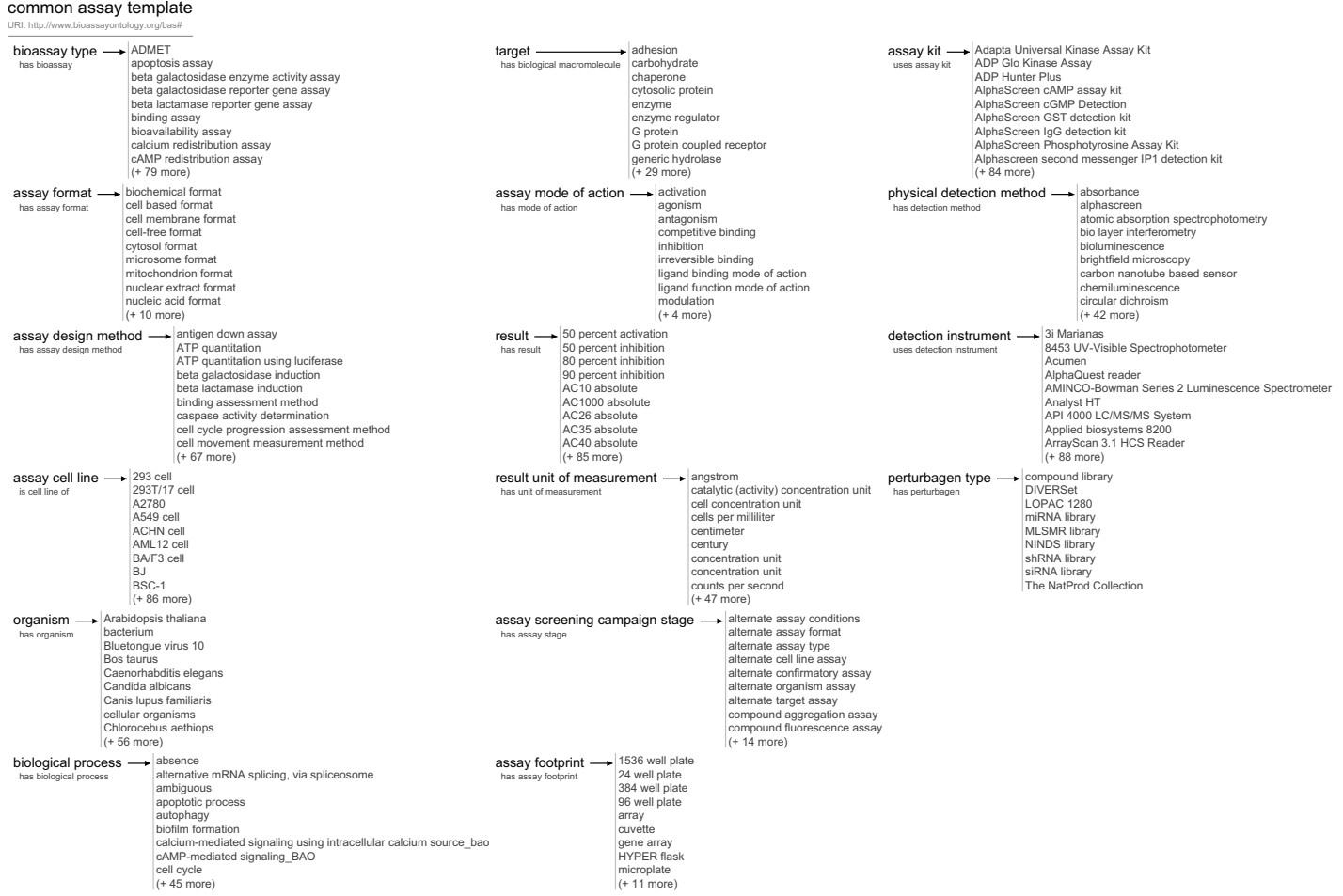

**Figure 2** An overview of the CAT at the time of publication.

and subgroups, our current template errs on the side of simplicity, and includes just 16 different assignments, each of which is associated directly with the top-level assay, and each of which has a list of associated values (examples shown in Fig. 2).

A template can be customized as necessary, and once it is ready, it can be used to define the way in which assays are annotated. The data model is designed to enable software to compose a user interface: presenting each of the categories, and making use of the selected values as the options that are made available to the user. It is essentially a way to restrict and simplify the large scope of the BAO, reduce the degrees of freedom, and remove ambiguity. Having curated the assignments and values so that the lists consist of the minimum number of relevant possibilities, each of them decorated by a meaningful label and a more detailed description, it becomes possible to design a user experience that is suitable for a scientist who is an expert in the field, but does not necessarily know anything about semantic web concepts.

In order to explore this approach, we have created a software package called the BioAssay Schema Editor, which is open source and available via GitHub. It is written using

Java 8, and runs on the major desktop platforms (Windows, Mac & Linux). The software implements the data model that we describe in this article.

Our priorities for this work are to: (1) establish a data model for bioassay templates; (2) create an intuitive software package for editing these templates and using them to annotate real data; and (3) collaboratively establish a CAT for general purpose use. We have put a considerable amount of effort into the user interface for editing templates, even though we expect only a small fraction of biologists will ever be directly involved in editing them. We have also invested significant effort towards developing a one-size-fits-most template, the CAT. Our goal with the CAT was to enable capture of ~80% of the most commonly used terms, and present them in a logical and concise way, so that a large proportion of users will be able to use it as-is to add a significant amount of value to their protocol data. In addition, the CAT can act as a starting point for modification if scientists would like to tailor the template.

Scientists working in research groups that routinely make use of terms that are not included in the CAT can elect to start with an existing template and add the missing assignments and values, and also delete whole groups of content that do not apply to their research. A research group may accumulate a collection of task-specific templates, allowing their scientists to pick the most appropriate one. By ensuring that the editor software is easy to use, runs on all platforms, and is open source, we hope to ensure that this option is quite practical for any research group with access to basic information technology expertise. We intend to encourage the community to make use of these resources, both as standalone tools and interoperating with the electronic lab notebook software that we are presently designing.

One of the implicit advantages of using semantic web technology as the underlying data format (triples), and a well established set of reference terms (the BAO and various linked ontologies), is that even if two scientists are annotating assays with different templates, it is highly likely that many or most of the terms will overlap, even if the templates were created from scratch. Since the final deliverable for an annotated assay is the semantic web, it means that the output can be subjected to the entire universe of software designed to work with RDF triple stores.[4] As more assays are annotated, the scope and power of queries and informatics approaches for enhancing drug discovery projects are similarly increased. With a large corpus of annotated assays available, scientists will be able to make better use of prior work for understanding structure activity relationships, uncovering experimental artifacts, building machine-learning models, and reducing duplicated efforts.

## METHODS

### Data model

The semantic description of templates and annotations uses a small number of additional URIs, each of which has the root stem http://bioassayontology.org/bat, and is denoted using the Turtle-style[5] abbreviated prefix "bat."

The hierarchical model for describing a template is shown in Fig. 3. Parent:child relationships denoted by an arrow indicate one-to-many relationships, while the

[4] See W3C Resource Description Framework: http://www.w3.org/RDF.

[5] See W3C RDF Turtle: http://www.w3.org/TR/turtle.

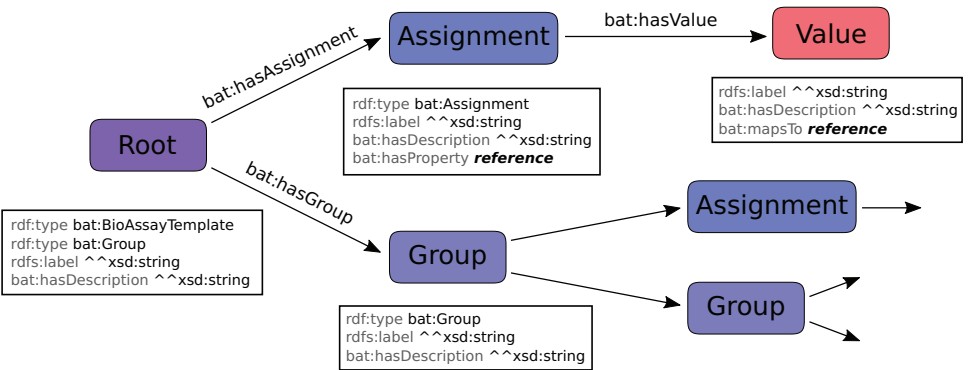

**Figure 3** BAT data model, which is used to describe a template.

properties listed in the boxes underneath the nodes are one-to-one relationships. A template definition begins with the *root*, which is distinguished by being of type bat: BioAssayTemplate. The root is also of type bat:Group, and has some number of child nodes, which are themselves either assignments or subgroups.

An assignment node has several scalar properties, including label and description, and it also refers to a *property* resource. These are typically mapped to URI resources found within the BAO (e.g. http://www.bioassayontology.org/bao#BAO_0000205, label: "has assay format"). Each assignment has some number of values associated with it, and these make up the list of available options. Each value is primarily identified by the resource that it maps to, which is typically found in the BAO (e.g. http://www.bioassayontology.org/bao#BAO_0000219, label: "cell based format"). Besides the label and description, which are customizable within the template data model, the reference URI has its own implied class hierarchy (e.g. "cell based format" is a subclass of "assay format"), which is not encoded in the template data model, but is inferred once it is paired with the BAO and its linked ontologies.

The schema for annotation of assays is shown in Fig. 4. The assay is given a distinct URI, and is associated with several properties such as label and description. The template is recorded, as is an optional reference to the origin of the assay (which may be a semantic web resource, or a DOI link to a journal article). The free-text description of the assay can also be recorded using the *hasParagraph* predicate.

The assay is associated with some number of annotations, which are primarily linked to assignments within the corresponding template. For annotations that assert a URI link, the *hasValue* predicate typically corresponds to one of the available values that was prescribed for the assignment in the template definition, and generally refers to a term defined in the BAO, though custom references can be used–or the annotation may be specified using the *hasLiteral* predicate instead, which means that the user has entered data in a different form, typically text or a numeric value. The *hasProperty* predicate is generally copied from the corresponding assignment.

When annotating an assay, each assignment may be used any number of times, i.e. zero instances means that it has been left blank, while asserting two or more triples means that

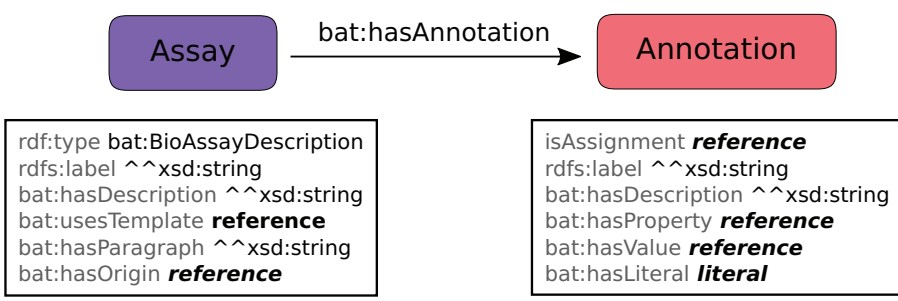

**Figure 4** **Data model for annotated assays, which is used to apply a template to a specific assay.**

all of the values apply. The relationship between assays and annotations has no nesting: the intrinsic group/sub-group structure of any particular annotation can be inferred from the template, since the *usesTemplate* and *isAssignment* predicates refer to the origins in the template.

## Software

The BioAssay Schema Editor is available from GitHub (https://github.com/cdd/bioassay-template) and may be used under the terms of the Gnu Public License 2.0.[6] The code is written using Java 8, and the user interface is based on JavaFX. Semantic web functionality is implemented by incorporating the Apache Jena library.[7] The project includes a snapshot of the BAO[8] and some of the linked ontologies, as well as the latest version of the CAT schema. It should be assumed that the project will continue to evolve until well after the publication date of this article.

The application operates on a datafile referred to as a *schema*, which is represented as a collection of triples (in Turtle format, with the extension .ttl). A schema is expected to include a single template, for which the root node is of type bat:BioAssayTemplate, and may optionally contain any number of assays that have been (or will be) annotated using that same template. Triples are used as the serialization format in order that the editable files can be used as-is by a Triple store, and become a part of the semantic web with no further modification.

Figure 5 shows the main window for the application, which has loaded a contemporary version of the *CAT*, and has several accompanying assays awaiting annotation. The components that make up the template are shown as a hierarchy on the left hand side of the panel. Selecting any of the groups or assignments causes the detail view on the right to be filled in with the corresponding content.

Adding, deleting, renaming etc. of groups, assignments and values is fairly mundane, and follows standard desktop user interface design patterns. Selecting URI values for properties and values requires a more specific interface, and is composed by summarizing the BAO vocabulary, which is loaded into the application at the beginning. Resources can be selected using a dialog box that can present the list of options in a flat list, with an optional search box for restricting the list (Fig. 6A) or by using the hierarchy view that shows the position in the BAO ontology (Fig. 6B).

[6] Gnu Public License 2.0: http://www.gnu.org/licenses/gpl-2.0.en.html: the license allows anyone to use the source code for any purpose, on the condition that products making use of it must be made available under a license that is at least as open. Copyright for the project is held by Collaborative Drug Discovery, Inc.

[7] See Apache Jena project: http://jena.apache.org.

[8] Downloadable OWL files for the BAO: http://bioassayontology.org/bioassayontology.

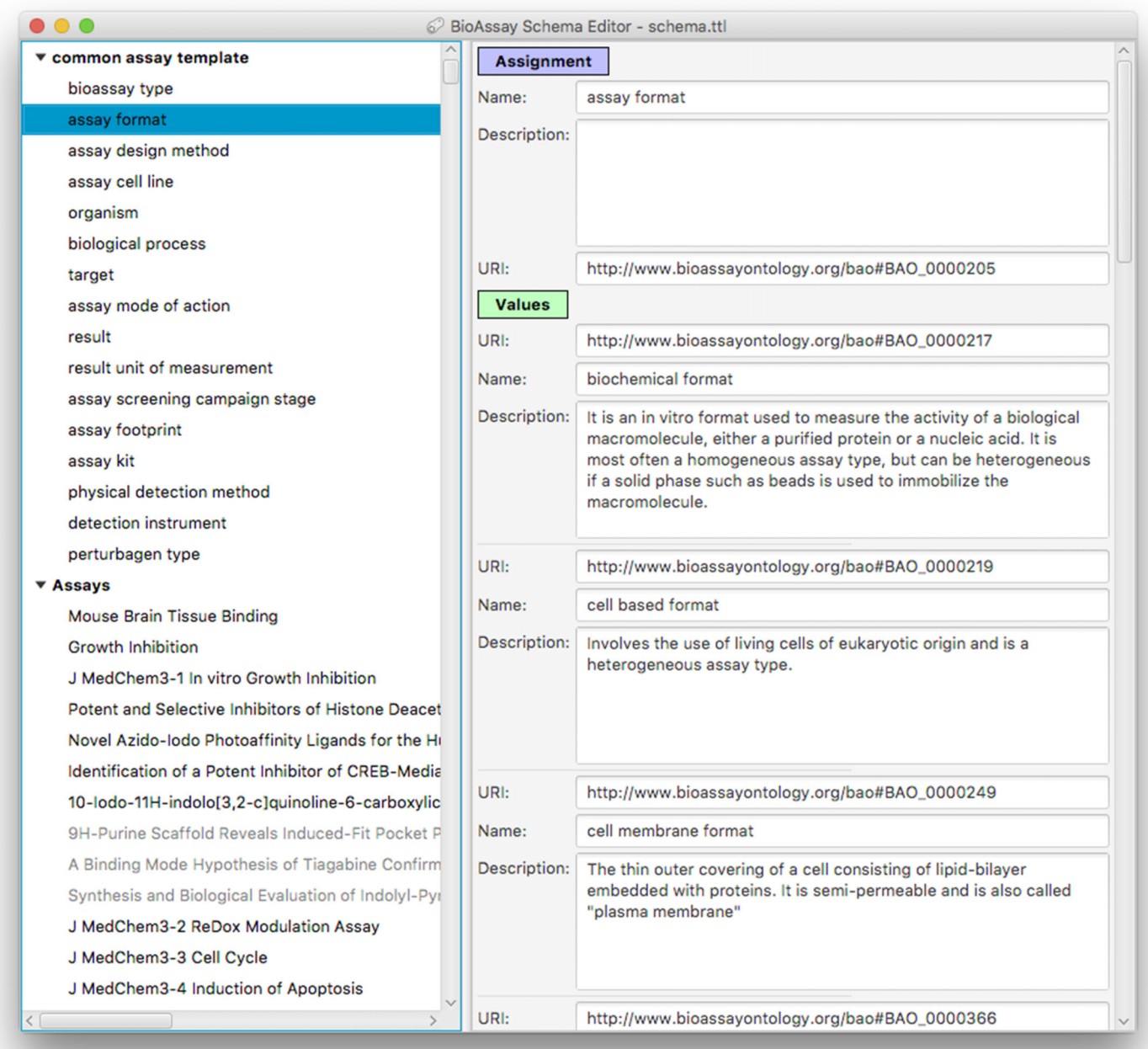

**Figure 5** **A snapshot of the BioAssay Schema Editor.** On the left hand side the current template is shown at the top (with its hierarchy of groups and assignments), and any assays currently in progress shown underneath. The panel on the right shows the details for an assignment–*assay format*–and the prescribed values that are associated with it.

The dialog box can also be used to add multiple values at once, which is particularly convenient when a branch of the BAO encompasses multiple terms that are all valid options. When a resource is selected, its label and description are imported from the BAO into the template: these values can be edited after the fact, but by default they are the same as in the underlying vocabulary.

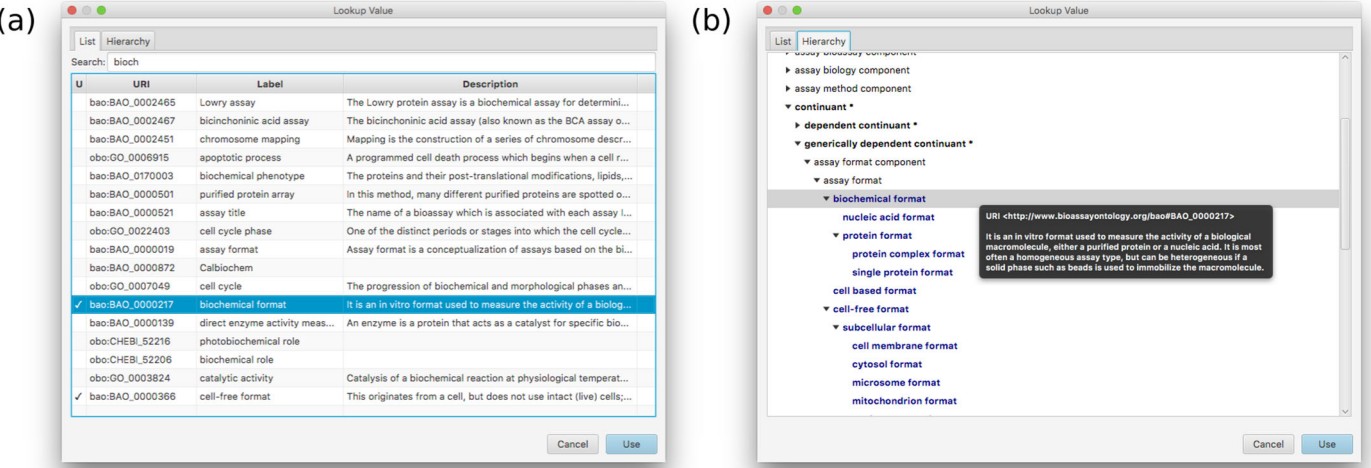

**Figure 6** **A snapshot of the two main tabs used for locating a value in the BAO.** (A) shows the list view, which is flat, while the (B) shows the values in context of the actual hierarchy of the underlying ontology.

The primary role of the schema editor is to provide a convenient way to edit templates, but in support of this goal, it also provides an interface to use the template to annotate assays. The interface can be used for generating training data (e.g. for model generation), but it is mainly intended as a way to 'test drive' the current template. Because the annotation process is directly derived from the template, having the two editing processes side by side is advantageous when the template is being designed. For example, the operator can begin annotating an assay, and if a value is missing from one of the assignments, or a new kind of assignment turns out to be necessary, this can be added to the template within the same editing session.

Figure 7A shows an example of an assay that has been annotated. The detail view has a placeholder for description text, which is particularly useful when the content has been imported from some external source, and the annotations are being made by converting the protocol text into semantic annotations. Clicking on any of the annotation buttons brings up a panel of options (Fig. 7B) that represent the prescribed values for the assignment. Each of the assignments can be left blank, annotated once, or given multiple values. The ideal use case is when the value (or values) occurs within the list of prescribed values, but since the data model allows any URI, the user interface also allows the user to insert a custom URI. In cases where no URI is listed in the template (e.g. a concept that does not have an established URI), it is possible to add plain text for any of the assignment annotations. While this has no meaning from a machine-learning point of view, it can serve as a convenient placeholder for terms that will be invented in the future.

## RESULTS

### Templates

We set out to create a CAT that includes the basic details essential to defining any bioassay: assay type, format, target and biology, results and pharmacology, and other details.

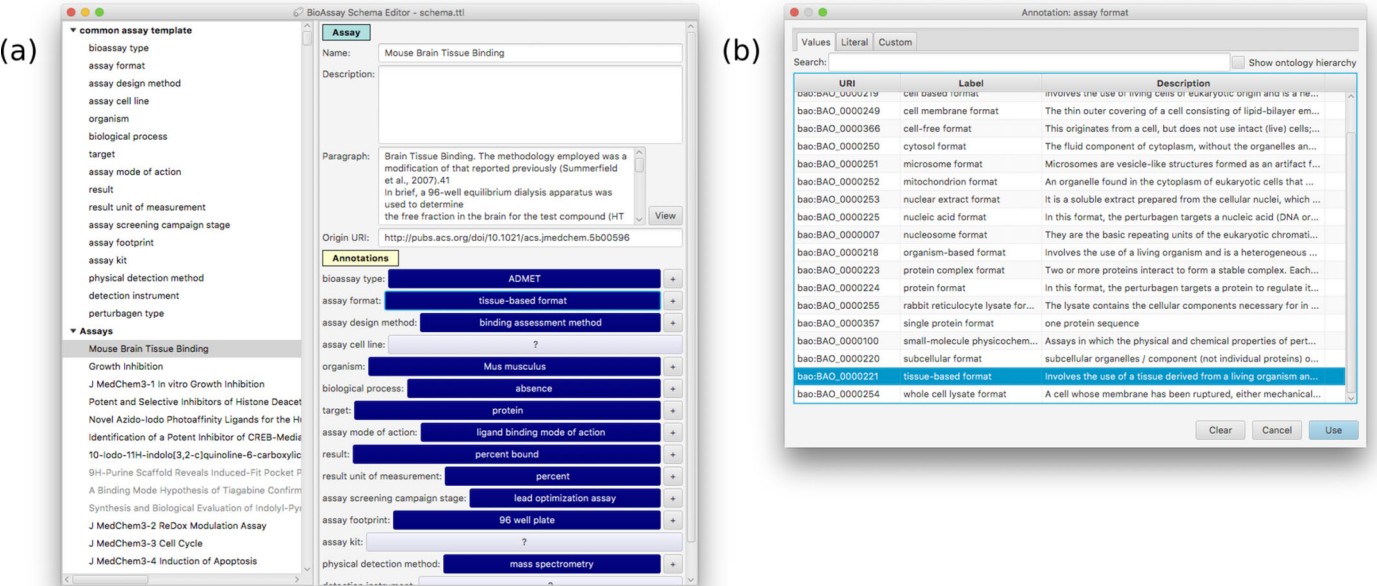

**Figure 7  A snapshot of the annotation interface that is available within the template editor.** (A) The current template can be applied to specific assays within the same overall user interface, which is a convenient way to evaluate its suitability. Selecting any of the assignments brings up a dialog box presenting all of the prescribed values (B).

The CAT was developed with the opposing goals of identifying assignments that (1) would be limited in number in order to be not overly burdensome vs. (2) comprehensively cover the majority of the information contained in written descriptions of bioassays. We also considered the type of information that would be utilized by an end user attempting to search, filter, and aggregate assays by their bioassay annotations. For example, details such as the assay footprint (plate type), assay kit, and detection instrument were included because they may be useful terms for identifying experimental artifacts. Biological process and other target-related information were included to enable aggregating results across similar drug discovery projects for model-building and other applications. Finally, we limited assignments to those where the BAO offered sufficient options for possible values. Since the goal of the project is to generate machine-readable assay annotations, we avoided assignments where BAO terms were not available, such as those characterizing in vivo assays, and especially assignments whose values would be very specific for each assay, such as negative and positive controls. These areas will be addressed in the future once the underlying vocabulary (BAO or otherwise) is available sufficient to expand the domain. Similarly, the CAT falls short of capturing detailed protocol steps. In its present incarnation, it cannot be considered as a complete replacement for the text that is typically used to describe an assay, though we do intend to pursue this level of detail in future work. For the present, we are primarily concerned with utilizing the rich vocabulary within the BAO to achieve maximum impact with minimum additional burden on the end user workflow.

To develop the CAT, we used the following process: first, biologists independently considered each of the terms available in the BAO and prioritized assignments for the CAT. Each assignment was associated with a number of possible values based on the BAO hierarchy. Then, quantitative and qualitative approaches were used to determine if the prioritized assignments included in the CAT were sufficient to fully describe most assays. For the quantitative approach, we assessed the set of 1066 PubChem bioassays (*Wang et al., 2014*) that were previously annotated by hand by BAO experts (*Schürer et al., 2011*). In that exercise, the BAO experts aimed to fully annotate each assay, capturing all applicable information for more than a hundred different categories or terms. If there was not an applicable value, the assignment or category was left blank. We analyzed the use of the BAO terms to assess the utility and comprehensiveness of the assignments included in the CAT compared to the remaining terms. We found that the 16 CAT assignments were annotated in 81% of the 1066 PubChem assays compared to 33% for the remaining terms. We also found that 95% of the values for CAT assignments were BAO terms rather than literal or non-URI based terms, compared to 63% in the remaining categories. These results suggested that the CAT includes assignments that are both relevant to the majority of assays as represented in PubChem and well covered by the BAO.

For an in-depth qualitative assessment of the CAT, biologists annotated a wide variety of assays, encompassing different assay types (e.g., cell viability, enzyme activity, binding, and ADMET), assay formats (e.g., cell-based, biochemical, microsome, organism, tissue, etc.), and assay design methods (e.g., ATP quantitation, cell number, immunoassays, gene expression, radioligand binding, etc.), as summarized in Table 1. We found that in many cases, both from assay descriptions available from PubChem and from in-house screening assay descriptions, the CAT captured much of the relevant information. For example, annotating an assay for cell viability (PubChem ID 427) shows that all but two of the 16 CAT assignments are readily annotated from the short descriptive information provided (Fig. 8). 'Target' is left blank, as it is not applicable (this assay aims solely to identify cytotoxic compounds); 'Detection Instrument' was not noted. Similarly, as shown in Fig. 9, all applicable CAT assignments (15 of the 16) are annotated from the description of a competitive binding assay (PubChem ID 440). Figure 9 also illustrates that multiple values can be annotated for a single assignment, enabling content from complex assays to be captured. Together, these two examples highlight that both cell-based and biochemical assays can be extremely well-suited to be annotated using the CAT.

However, there were some cases where the CAT was less effective in capturing important information. For example, 14 of the 16 CAT assignments could be annotated for PubChem ID 488847, some with multiple values; however, the 'big picture' view of this rather complex primary assay is not as readily apparent from its 'CAT profile' as from a single sentence in the description (Fig. 10). In addition, this PubChem record had extensive technical details such as reagent components, liquid handling volumes and instruments, times of incubation and plate processing steps, which could be important for identifying matching assays or interpreting the results. Another example of a poor fit for the CAT, as noted earlier, are in vivo assays. These are largely beyond the scope of this

**Table 1  Representation of CAT in Sample Assay Set.**

| CAT assignment | Test assays (of 43) with at least 1 value | # Of unique values annotated |
|---|---|---|
| Bioassay type | 43 (100%) | 24 of 88 |
| Assay format | 43 (100%) | 6 of 19 |
| Assay design method | 43 (100%) | 20 of 76 |
| Assay cell line | 24 (55.8%) | 15 of 95 |
| Organism | 41 (95.3%) | 11 of 65 |
| Biological process | 40 (93.0%) | 28 of 54 |
| Target | 32 (74.4%) | 13 of 38 |
| Assay mode of action | 43 (100%) | 8 of 13 |
| Result | 41 (100%) | 16 of 94 |
| Result unit of measurement | 32 (74.4%) | 6 of 56 |
| Assay screening campaign stage | 40 (93.0%) | 8 of 23 |
| Assay footprint | 36 (83.7%) | 5 of 20 |
| Assay kit | 9 (20.9%) | 5 of 93 |
| Physical detection method | 42 (97.7%) | 11 of 51 |
| Detection instrument | 26 (60.5%) | 9 of 97 |
| Perturbagen type | 20 (46.5%) | 3 of 9 |

(a) We have developed a 1536-well cell-based assay for quantitative high throughput screening (qHTS) against a number of cell lines to determine in vitro cytotoxicity of small molecules. This particular assay uses the Hek 293 cell line which is derived from human embryonic kidney cells (transformed with adenovirus). The CellTiter-Glo luminescent cell viability assay (Promega) is a homogeneous method to measure the number of viable cells in culture. The end point readout of this assay is based on quantitation of intracellular ATP, an indicator of metabolic activity, using the luciferase reaction. Luciferase catalyzes the oxidation of beetle Luciferin to oxyluciferin and light in the presence of ATP. The luminescent signal is proportional to amount of ATP present. Using the CellTiter-Glo luminescent cell viability assay, the amount of cellular ATP was measured in the Hek293 cell line with complete culture medium following compound treatment for 40 hours. The assay was performed in opaque white Kalypsys 1536-well plates. In the screen, tamoxifen and doxorubicin were used as positive controls. Library compounds were measured for their ability to cause acute toxicity in the cell line, as reflected by a decrease in intracellular ATP levels, in a concentration-dependent manner. Data were normalized to the controls for basal activity (DMSO only) and 100% inhibition (100 uM tamoxifen). AC50 values were determined from concentration-response data modeled with the standard Hill equation.

**Key**
Annotated with URI
Added as literal
Not annotated: missed opportunity
Requires more advanced template model

(b) PubChem Assay (ID 427)
Origin: http://pubchem.ncbi.nlm.nih.gov/bioassay/427

bioassay type —has bioassay→ cell viability assay
assay format —has assay format→ cell based format
assay design method —has assay design method→ ATP quantitation using luciferase
assay cell line —is cell line of→ HEK293
organism —has organism→ Homo sapiens
biological process —has biological process→ cell death
target —has biological macromolecule→ (not assigned)
assay mode of action —has mode of action→ modulation
result —has result→ AC50
result unit of measurement —has unit of measurement→ (not assigned)
assay screening campaign stage —has assay stage→ primary assay
assay footprint —has assay footprint→ 1536 well plate
assay kit —uses assay kit→ CellTiter-Glo Luminescent Cell Viability Assay
physical detection method —has detection method→ luminescence method
detection instrument —uses detection instrument→ (not assigned)
perturbagen type —has perturbagen→ compound library

**Figure 8  First example of PubChem Assay text ideally suited for annotation with the CAT.** (A) Text from description in PubChem Assay ID 427: yellow = information captured in CAT, green = information not captured but possible for a future version (e.g., controls, data processing), red = information beyond the scope of BAO (technical details). (B) CAT assignments in BioAssay Schema Editor.

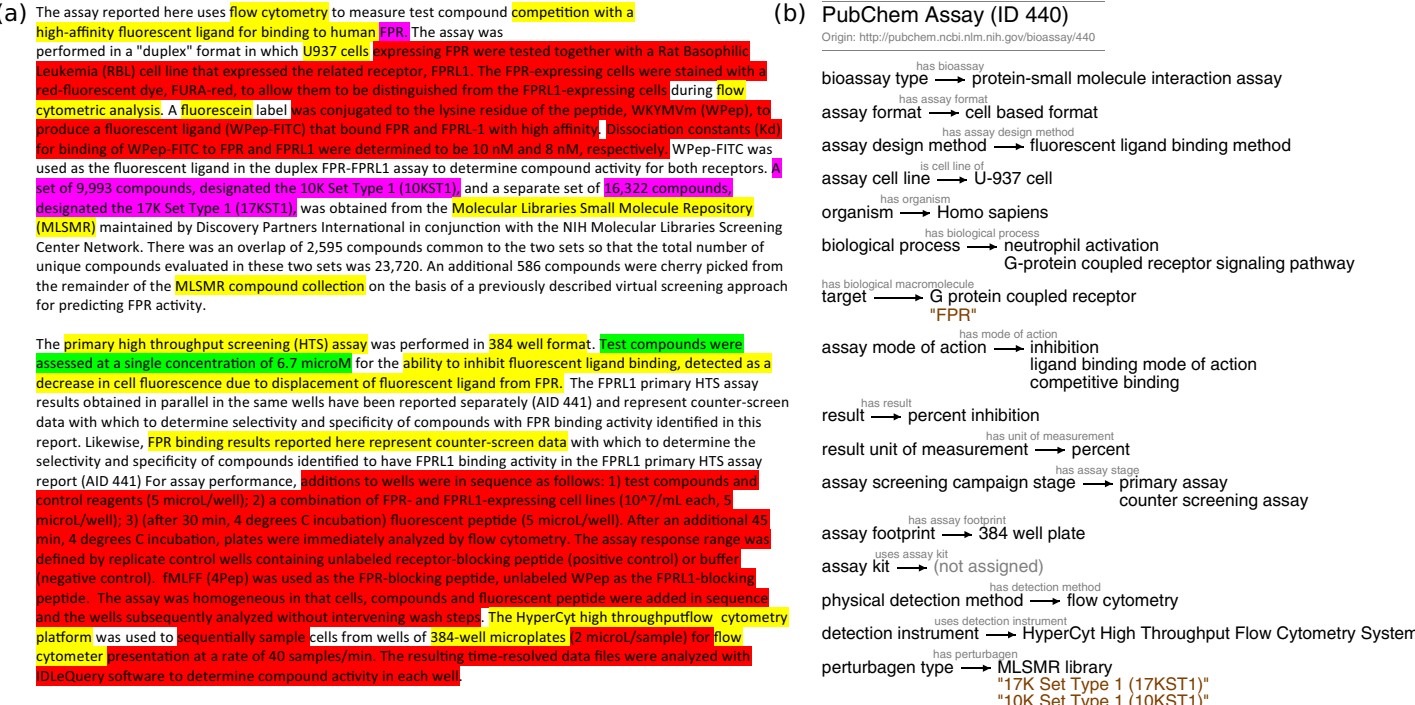

**Figure 9 Second example of PubChem Assay text ideally suited for annotation with the CAT.** (A) Text from description in PubChem Assay ID 440: yellow = information captured in CAT, pink = information added as 'literal' values (i.e., too specific to exist as a BAO entry, but deemed valuable), green = information not captured but possible for a future version (e.g., controls, data processing), red= information beyond the scope of BAO (technical details). (B) CAT assignments in BioAssay Schema Editor. Annotations added as 'literal' values are highlighted yellow and contained in single quotes. Note that multiple values for a single CAT assignment can be annotated (*target biological process, assay mode of action, assay screening campaign stage, perturbagen type*).

effort, which is currently constrained to terms defined by the BAO: key parameters such as route of administration, dose, dose units, type of model (e.g. xenograft, disease) are not well represented. These and other limitations will be addressed in the future by adding or extending the underlying ontologies.

Finally, as noted earlier, we designed the CAT to be a 'one-size-fits-most' template. A summary of assignments for the complete set of assays annotated in the course of developing the CAT shows we have achieved this (Table 1). One consequence of this 'one-size-fits-most' strategy is that certain attributes (such as those highlighted in green or red in Figs. 8 and 9) have been omitted. Depending on one's perspective, these types of data (such as positive and negative controls, data processing/normalization steps, relevant disease indication, and specific protocol details such as pre-incubation of compounds with the target, time or temperature of an assay) could be viewed as essential. We decided to exclude this type of information from the CAT because of irregularity of appearance in bioassay descriptions, the lack of coverage by the BAO, or incompatibility with the current data model. Expanding into this area is an opportunity for future development, and it should be noted that the CAT may be used as a starting point for templates that provide a set of assignment options that are customized for subcategories of assays,

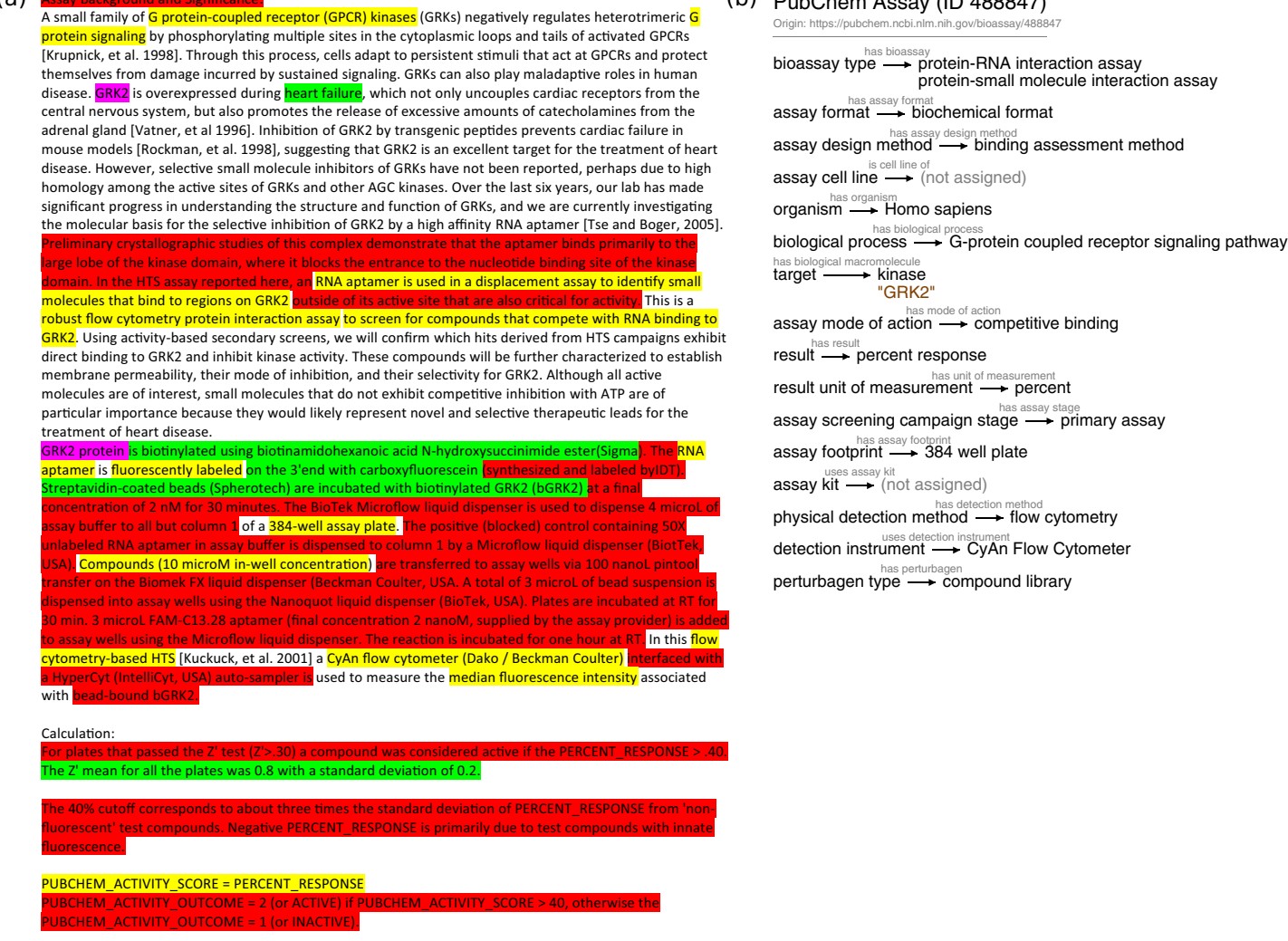

**Figure 10** **Example of an assay partially suited for annotation with the CAT.** (A) Text from description in PubChem Assay ID 488847: yellow = information captured in CAT, pink= information added as 'literal' values (i.e., too specific to exist as a BAO entry, but deemed valuable), green = information not captured but possible for a future version (e.g., controls, labels of target and ligand, assay quality data (Z′)), red = information beyond the scope of BAO (technical details). (B) CAT values assigned in the BioAssay Schema Editor capture key parameters of the assay yet do not capture the complexity of the assay articulated in the single sentence (arrow): "a flow cytometry protein interaction assay to screen for compounds that compete with RNA binding to GRK2."

or even specific projects. We believe the next immediate step should be to apply our CAT to a large (> 10,000) set of assays, both to facilitate new meta-analyses and to identify potential gaps in annotation revealed by such studies.

## PubChem

Possibly the most voluminous source of openly accessible bioassay data can be found on PubChem, which hosts more than 1.1 million assay records at the time of publication, and is growing rapidly. These are individually associated with the chemical structures of the compounds for which the measurements were made. Each of the assays is decorated

with several descriptive fields that are essentially plain text, and which are populated by contributors during the upload process, or in some cases by an import script transferring data from other sources. While many of the entries contain a significant amount of detail, the phrasing style and level of detail varies considerably, often erring on the side of too little or too much information about the assay protocol.

Nonetheless, the PubChem assay collection represents one of the best and most convenient sources of data for annotation purposes, and for this reason we have added a feature to the BAT editor that explicitly searches for PubChem records, as shown in Fig. 11.

The dialog box allows the user to type in a PubChem Assay ID number, or to hit the button labelled *Random*, which picks an arbitrary assay from the entire collection, and fills in the corresponding text and URI of origin. While a large proportion of assays loaded into PubChem contain only sparse tags about the data source, or the abstract of the corresponding publication, there are a significant number of records that contain lengthy descriptions of the assay. The dialog box provides an opportunity for the user to tidy up the text (e.g. removing irrelevant content) prior to importing it into the schema. The content is then added to the list of assays being annotated within the schema model, whereby the origin is recorded as a link to the assay, and the text is associated using the *hasParagraph* predicate. Once the text is augmented with annotations using the current template, it becomes a useful entry for training data. This is one of our main strategies for generating a corpus of data for machine-learning purposes, which will ultimately find its way into a user friendly ELN for bioassay annotation.

## Analysis

Because the data model we describe is based on semantic web triples, and the file format that is used by the BioAssay Schema Editor is made up of triples (in Turtle format), it means that any templates and assay annotations can be loaded directly into a triple store database, and queried using SPARQL queries. Content can be hosted on private servers for local use, or it can be exposed to the greater web of connected data. Supplementary Information 1 describes a configuration script for the open source Apache Fuseki Jena server which can be used to load the BAO, its related ontologies, and some number of files saved with the BioAssay Schema Editor, which can then be served up as read-only content.

Once the content is available via a SPARQL endpoint, there are a number of boilerplate queries that can be used to extract summary and specific information. Fetching a list of all bioassay templates can be accomplished using the following query:

```
PREFIX bat: <http://www.bioassayontology.org/bat#>
PREFIX rdfs: <http://www.w3.org/2000/01/rdf-schema#>
SELECT ?template ?label ?descr WHERE
{
    ?template a bat:BioAssayTemplate ; rdfs:label ?label .
    OPTIONAL {?template bat:hasDescription ?descr .}
}
```

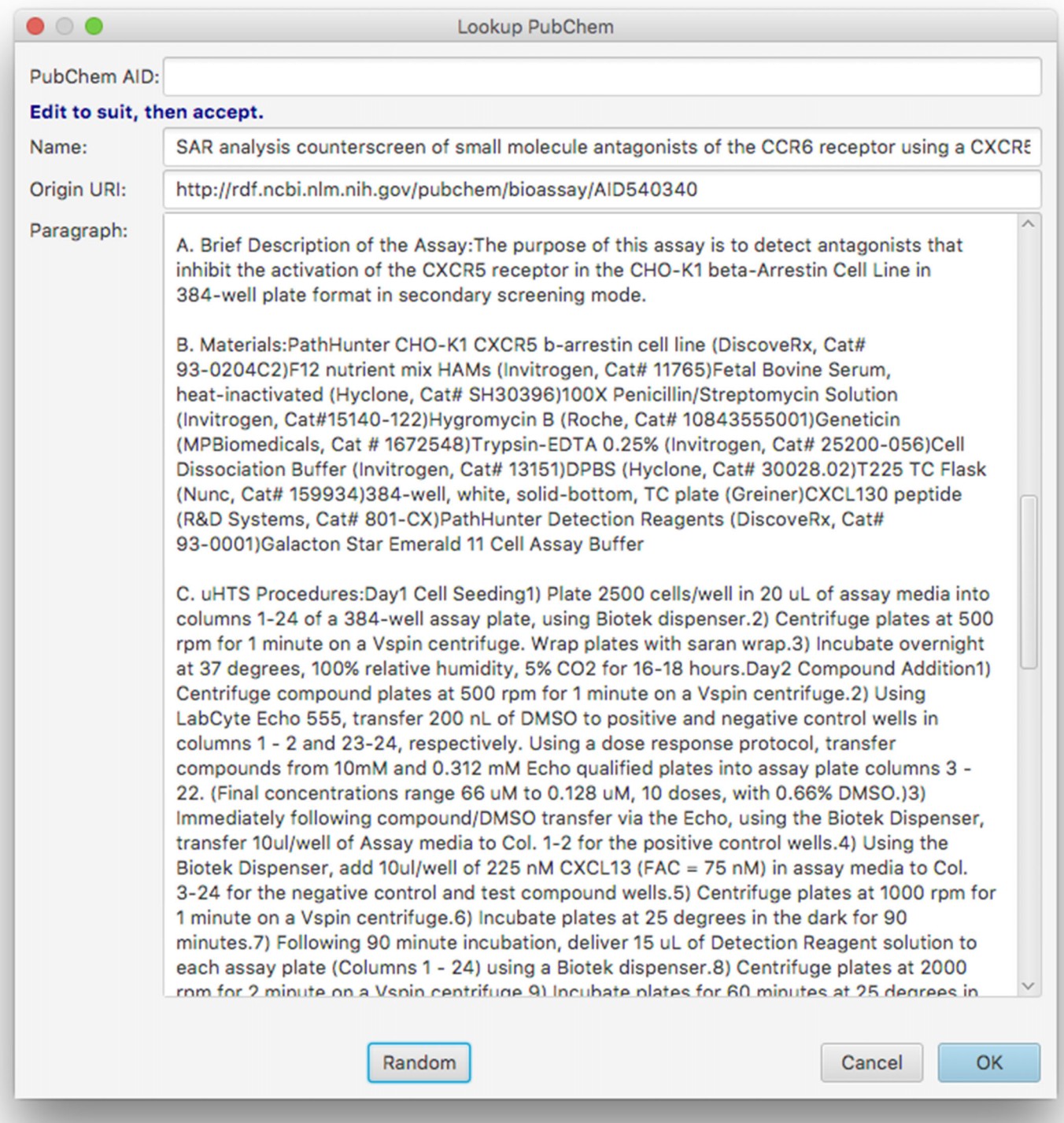

**Figure 11** Dialog box for random lookup of assays from PubChem.

The above query identifies any resource that is tagged as having the *BAT* type. Obtaining information about the assignments that are associated with a template can be done by looking for resources of type *Group* that are associated with it. Obtaining

a summary list of assignments that are attached to the top level (i.e. not within a subgroup) can be accomplished with a query similar to the following (using the same prefixes as above) which explicitly references the CAT:

```
SELECT ?assn ?label ?descr ?property ?numValues
{
    <http://www.bioassayontology.org/bas#CommonAssayTemplate>
                    bat:hasAssignment  ?assn .
    ?assn a bat:Assignment ;
        rdfs:label ?label ;
        bat:hasProperty ?property .
    OPTIONAL {?assn bat:hasDescription ?descr .}
    {
        SELECT ?assn (COUNT(?value) as ?numValues) WHERE
        {
            ?assn bat:hasValue ?value .
        }
        GROUP BY ?assn
    }
}
ORDER BY ?label
```

Similarly, assignments with one level of nesting can be obtained with a slightly longer query, which explicitly inserts a subgroup in between the template and assignment:

```
SELECT ?group ?glabel ?assn ?label ?descr ?property ?numValues
{
    <http://www.bioassayontology.org/bas#CommonAssayTemplate>
                    bat:hasGroup ?group .
    ?group a bat:Group ;
        rdfs:label ?glabel ;
        bat:hasAssignment ?assn .
    ?assn a bat:Assignment ;
        rdfs:label ?label ;
        bat:hasProperty ?property .
    {
        SELECT ?assn (COUNT(?value) as ?numValues) WHERE
        {
            ?assn bat :hasValue ?value .
        }
        GROUP BY ?assn
    }
}
ORDER BY ?glabel ?label
```

To query for information about the prescribed values for assignment (in this case the bioassay assignment from the CAT), the following query can be used:

```
SELECT ?property ?value ?label
{
    <http://www.bioassayontology.org/bas#Bioassay>
        bat:hasProperty ?property ;
        bat:hasValue
        [
            bat:mapsTo ?value ;
            rdfs:label ?label
        ] .
}
```

The query specifically pulls out the *property* field, which is typically a link into the BAO property terms, and the *value* field, which is typically a link into the BAO classes. Pursuing either of these resources provides a wealth of implicit information, partly from the hierarchical nature of the BAO terms, and the unlimited opportunities for these terms to be linked to other semantic resources.

To obtain a list of assays that have been annotated using one of the templates, the following query can be used:

```
SELECT ?assay ?label ?descr ?template WHERE
{
    ?assay a bat:BioAssayDescription ;
        rdfs:label ?label ;
        bat:usesTemplate ?template .
    OPTIONAL {?assay bat:hasDescription ?descr .}
}
```

Obtaining all of the annotations for such an assay can be done with:

```
SELECT ?assn ?label ?property ?value ?literal ?group WHERE
{
    <http://www.bioassayontology.org/bas#ExampleAssay>
                    bat:hasAnnotation ?annot .

    ?annot bat:isAssignment ?assn ;
        rdfs:label ?label ;
        bat:hasProperty ?property .
    OPTIONAL {?annot bat:hasValue ?value}
    OPTIONAL {?annot bat:hasLiteral ?literal}
    ?group a bat:Group ; bat:hasAssignment ?assn .
}
```

Because annotations are directly attached to an assay description, hierarchical information about the nature of the assignment can be obtained by further investigating the template definition of the assignment (*?assn*) or either of the linked BAO terms (*?property* and *?value*).

## CONCLUSION

We have developed a data model and interactive tool that can be used to narrow the degrees of freedom from the BAO and its linked dependencies. This has been done in order to facilitate content creation activities, so that semantic annotation of assay protocols can be carried out by a domain expert with no corresponding expertise with the underlying ontology. We have provided a proof of concept tool that creates a user interface based on the template data model, and made this available to the community as open source.

The data model that we have created follows a simplistic pattern, where elementary facts can be asserted. By leveraging the implied value of the underlying ontology, a small collection of a dozen or so such annotations provides a significant amount of machine-readable context about the assay. While insufficient to completely define an assay protocol experiment, this stands in contrast to the standard practice of providing essentially zero machine-readable information (i.e. plain English text with quasi-standardized jargon).

We have made available the CAT which was designed by biologists with the objective of leveraging the BAO to provide the largest amount of useful, relevant, machine-readable information with the fewest number of additional data points needing to be captured by the originating scientist. The CAT is expected to be useful for a wide variety of sorting, filtering, and data aggregating tasks that drug discovery scientists need to be able to carry out on a large scale, but currently cannot due to the absence of machine-readable annotations.

The CAT prioritizes 16 assignments that biologists consider most central to describing their assays and reporting assay results. Annotations for these assignments will enable biologists to ask complex queries. For example, one could ask if there are systematic differences in cell-based versus biochemical-based assays for a certain target class, such as kinases. One could determine if a certain assay set-up, such as 96-well plates using a spectrophotometer were likely to have a higher hit rate. Similarly, one could identify if a certain compound or class of compounds is active in multiple assays, and if those assays assess similar biological processes or if the activity is likely to be an artifact.

By focusing on 16 assignments out of more than a hundred options available in the BAO, the CAT is meant to impose a minimal burden for annotating scientists. Our goal is to make annotating assays simple and easy so that the practice may be generally adopted. Templates are malleable and scientists can easily include other assignments.

One critical type of information that is not included in the current framework is protocol steps, which would be essential for directly comparing two assays. In the future, it would be useful if this information were machine-readable. However, semantic technology using a simplistic data model like the BAT cannot capture sequences of information. Capturing procedural or protocol steps would require the development of a

more complex data model. Under the current system, we imagine that queries using annotations from the CAT will allow scientists to hone in on similar assays, but for the moment, experts will still need to read the full assay descriptions to make decisions about combining different assays' data sets.

We have carried out this work in the context of a much larger scope, which is to provide scientists with tools to easily annotate bioassays and other related experiments in a way that is complete and machine-readable. Given that the standard industry practice does not involve adding any machine readable data to assay protocols, and that there are currently no widely available tools to do so with a user experience that is sufficiently painless for mass adoption, we have taken an incremental approach. This additional work has been done in order that we can continue with our previous work that was focused on using machine learning techniques to accelerate manual assignment of assays (*Williams et al., 2012*). Our immediate follow-up goals are to make use of the CAT to gather a large corpus of training data, both from active users of CDD Vault, and from existing repositories such as PubChem. This training data will be used to ensure that our enterprise ELN tools will be supported by machine learning technology as soon as they are unveiled.

We are also pursuing options for extending the BAT data model so that it is capable of capturing more sophisticated information about assays, e.g. linking to other ontologies to cover more types of assays; adding terminology for capturing quantities; addition of indefinite numbers of preparation steps; dependent assignment types, etc. One critical step when we enable connecting with other ontologies will be the ability to link the 'Target' to a unique identifier such as geneid or UniProtID. Each unique target identifier can be associated with a rich array of corresponding GO terms, of which a subset are mapped into the default selection of BAO classes. This will enable comparison of assays based on specific targets and related biological processes or molecular functions. While our first objective is *horizontal* scaling, i.e. ensuring that all assay protocols have semantic annotations that make a large portion of the content machine-readable, pursuing *vertical* scaling is also of great interest, i.e. making it possible for the semantic annotations to replace the need for use of English text (*Soldatova et al., 2014*). This brings about some exciting possibilities beyond just improvement of searching and matching, such as uploading protocols to robotic assay machinery, or making the publication process multi-lingual, thus alleviating a considerable burden to non-native English speakers. Pursuing this goal will require significant additions to the BAO itself, as well as making increased use of borrowed terms from other ontologies.

The technology that we have described in this article has been created for the purpose of improving the electronic lab notebook (ELN) technology that is offered by Collaborative Drug Discovery, Inc. (CDD), and we have begun work on a web-based interface for using templates such as the CAT for annotating assay protocols.[9] We have disclosed all of the underlying methods, data and open source code because we welcome participation by anyone and everyone. While CDD is a privately held for-profit company, it is our firm belief that improvement to this particular aspect of scientific research is a positive sum game, and we have more to gain by sharing than by keeping our technology entirely proprietary.

[9] A preliminary version of the web interface can be found at http://bioassayexpress.com. At the time of writing this service is in an early pre-alpha phase, but will be updated as the project progresses.

## SUPPORTING MATERIALS

The BioAssay Schema Editor is publicly available from GitHub (https://github.com/cdd/bioassay-template). The source code for the application is available under the terms of the Gnu Public License (GPL) v2, which requires that derived works must also be similarly open. The underlying semantic data model for the template and assay annotation, as well as the CAT, are public domain: they are not copyrighted, and no restrictions are placed on their use. The BAO is available from the corresponding site (http://bioassayontology.org/bioassayontology) under the Creative Commons Attribution License v3.

### Funding

This work was funded in part from the NIH NCATS Phase 2 SBIR Grant # 2R44TR000185-02 "Simplifying Encoding of Bioassays to Accelerate Drug Discovery" as described on https://projectreporter.nih.gov/reporter.cfm. The funders had no role in study design, data collection and analysis, decision to publish, or preparation of the manuscript.

### Grant Disclosures

The following grant information was disclosed by the authors:
NIH NCATS Phase 2 SBIR: 2R44TR000185-02.

### Competing Interests

All authors are employees of Collaborative Drug Discovery, Inc.

### Author Contributions

- Alex M. Clark conceived and designed the experiments, performed the experiments, contributed reagents/materials/analysis tools, wrote the paper, prepared figures and/or tables, performed the computation work, reviewed drafts of the paper.
- Nadia K. Litterman conceived and designed the experiments, analyzed the data, contributed reagents/materials/analysis tools, wrote the paper, prepared figures and/or tables, reviewed drafts of the paper.
- Janice E. Kranz conceived and designed the experiments, analyzed the data, contributed reagents/materials/analysis tools, wrote the paper, prepared figures and/or tables, reviewed drafts of the paper.
- Peter Gund analyzed the data, contributed reagents/materials/analysis tools, wrote the paper, reviewed drafts of the paper.
- Kellan Gregory contributed reagents/materials/analysis tools, wrote the paper, reviewed drafts of the paper.
- Barry A. Bunin contributed reagents/materials/analysis tools, wrote the paper, reviewed drafts of the paper.

## Data Deposition

GitHub: https://github.com/cdd/bioassay-template.

## Supplemental Information

Supplemental information for this article can be found online at http://dx.doi.org/10.7717/peerj-cs.61#supplemental-information.

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
