# Peer review of "BioAssay templates for the semantic web"

_PeerJ Computer Science, doi:10.7717/peerj-cs.61_

## Round 0.1 · original submission · Minor Revisions

Two reviewers suggested minor revisions, while the third suggested major. As the third author's comments are primarily about the characterization and interpretation of the problem being solved and the organization of the manuscript, I believe that these concerns can be addressed without requiring a fundamental reworking of the manuscript. Please address all comments, paying particular attention to feedback from reviewer 3.

·

Basic reporting

This paper describes the use of the BioAssay Ontology for describing screening assays. The authors make good points about the difficulty in using and applying biomedical ontologies to scientific workflows. To address this, they developed a software tool which contains a common assay template and is modifiable to suit user's needs. This type of software is valuable for researchers and will help aid in structuring scientific data and ensuring reproducibility.

Figures 1 and 2 are cumbersome to view. I think these could be removed and the authors could point to BioPortal or the OWL file to convey this information to the reader.

The screenshots are low resolution.

Experimental design

For your bioassay template, you need to include detailed instructions on your GitHub page or in the paper on how to access and use the CAT, it was not obvious to me how to open and use the software, and I don't find it very intuitive to use.

Validity of the findings

The goal of this software application is to enable researchers to capture semantic data about their research findings. However, there needs to be detailed instructions on how to use this software tool, as most basic science researchers are not familiar with how to access software via GitHub. I think this could be addressed in the paper and/or the ReadMe file in GitHub.

Additional comments

Add URL in abstract

Line 34: Should define what you mean by screening biologists

Line 38: Do you mean in publications or in the lab notebook, or both?

Line 41: Please clarify here - you are referring to the Materials and Methods section of a publication?

Line 58: Add link to BAO url

Line 92: The OWL file can also be viewed in Protege

Line 103: BioPortal says you have 3340 classes

Line 106: what is the BARD effort?

Line 143: Provide a URL for the GitHub site

Line 254: Should be 'detailed', not 'detail' in "The detail view has a placeholder..."

General comments:
In the BAO, there is a typo in the comment in the class: BAO_0003110 HMS, I think it should be 'Harvard' instead of 'Harward'. Seems like this should be an 'alternative term' instead of a comment, as well.

Where can users make term requests or submit issues to the BAO?

·

Basic reporting

This report is well written and addresses an important bottleneck in the use of domain ontologies to annotate biological data. The manuscript is well organized with clear introduction, methods, results and conclusions. Figure are adequate (some content needs to be improved / corrected). The developed BAT, CAT and software code with examples are made available open source.

Experimental design

The development of annotation software tools is an important area of research and development in the larger area of Semantic Web Technologies and Linked Data in particular in biomedical basic research where complex experiments, technologies and datatypes are common place. Authors report a simple approach to enable scientists who perform experiments and have little or knowledge of ontologies to make at least some basic use of the BioAssay Ontology to annotate biological assays (here primarily biochemical and cell-based assays)

Validity of the findings

The and methods approach are straight-forward, although I have not reviewed the code. I am very familiar with the BAO (disclosed above) and can therefore judge that the selection of the major category and their implementation are useful and cover basic annotations of straight-forward assays. It does not address a number of more complex annotations as the authors themselves indicate; these could be added in later as the tools evolve. BAO also provides the terminology and modeling to relate assays to each other (for example to describe a screening campaign), for example confirmatory, counter, alternate confirmatory, etc; this an important consideration outside the current solution. It could be mentioned. For reference, we have explained such assay relations and how they can be used to aggregate data in PMID 24078711.

Additional comments

While the content appears overwhelmingly correct, it could be pointed out that the majority over one million assays in PubChem are imported from other resources. This is relevant, because these are not strictly assays, but (for example in case of ChEMBL records) curated descriptions from the literature; they would typically describe the final results / outcome of a study, but not each step on the way (e.g. the screening campaign). While it does make sense to annotate these records using formal semantics, BAO allows to describe screening assays technologically and to some extent operationally, e.g. screening pipelines, multiplexes and parallelized assays. The best example therefore to point out in PubChem are the ones generated in the NIH funded Molecular Libraries Program, specifically MLSCN and MLPCN programs.
In the introduction the complexity of BAO is pointed out and exemplified with figures 1 and 2. Figure 1 appears to include many object properties that are not part of BAO and that are also not used in any assay descriptions in BAO. Many of those appear RO and BFO2.0 relations; some others appear to be made up, e.g. "has other".
BAO imports a module of relationships from RO / BFO, which can be viewed in an ontology editor. This has to be corrected and I would also recommend to indicate the source of the relations (BAO, RO, OBI, etc), because this is an important feature of compatibility. The same should be done for figure 2 (class hierarchy). BAO imports modules from many other ontologies (GO, ChEBI, CLO, EFI, DO, etc) and this should mentioned; perhaps where it is suggested that GO could be used to further annotate biological processes (74-79).

·

Basic reporting

The manuscript presents a common assay template (CAT) based on the previously developed ontology BAO and software (the BioAssay Editor) to ease the annotation of bioassays with formally defined BAO labels.

While the presented work on CAT is solid, with the potential users being involved into the development of CAT, there is no sufficient information on how the user requirements for the developed softwarewere identified and how the Editor was evaluated.

The overall logic of the submitted manuscript is not as crisp as one would expect:
- The authors state that there is a problem (“the major problem”) – a comparison of protocols.
- They claim that BAO can solve this problem – “in order to address this problem, the BAO was devised.
- “the BAO remains largely unused”.
- The presented in the manuscript work will make it easier to use BAO for the annotation of bioassays.

My concerns are:
- In my view, BAO is not sufficient to solve the problem of accurate comparison of the protocols. The granularity for the representation of experiments is not detailed enough, with some essential for the comparison information still missing in BAO. It can contribute to the solution of the problem, but no one evaluated to what degree. The authors need to make this point clear (since it is about the main problem discussed in the manuscript): provide convincing evidence that bioassay protocols annotated with BAO terms can be accurately identified as “identical, similar, or completely different”, or that BAO offers a partial solution to the problem, that BAO annotation can help with such identification.
- The manuscript gives an impression (possibly unintentional) that the main drive for the reported work is to make BAO more usable. Instead the motivation should be to solve the stated problem (or contribute to its solution).
- The conclusions of the manuscript do not refer to the stated problem, how the reported results solved the problem. Instead the conclusion section reports on the solution of another (even if related) problem.

The suggestion is to make the flow of the manuscript more logically coherent with a problem clearly stated, solutions to this problem described and evaluated, and conclusions about the considered problem made.

Experimental design

The process for the development of CAT is well described. 16 CAT were annotated in 81% of the 1066 PubMed assays.
Unfortunately there is no sufficient description of what methodology was used for the software development and how the quality and usability of the Editor was evaluated.

Validity of the findings

The findings are valid and of value to the research community. However it has not been estimated how the findings contribute to the solution of the stated problem.

Additional comments

The reported work is important and deserves to be published, but it needs to be better presented.

---

## Round 0.2 · accepted · Accept

Thanks very much for your revision addressing the concerns of the reviewers.